# Can Health-Promoting Schools Contribute to Better Health Behaviors? Physical Activity, Sedentary Behavior, and Dietary Habits among Israeli Adolescents

**DOI:** 10.3390/ijerph18031183

**Published:** 2021-01-29

**Authors:** Hila Beck, Riki Tesler, Sharon Barak, Daniel Sender Moran, Adilson Marques, Yossi Harel Fisch

**Affiliations:** 1Department of Health System Management, Faculty of Health Science, Ariel University, Ariel 407000, Israel; danielm@ariel.ac.il; 2Program in Gerontology, Faculty of Health Sciences, Ben-Gurion University of the Negev, Beer Sheeva 8499000, Israel; sharoni.baraki@gmail.com; 3Department of Pediatric Rehabilitation, The Edmond and Lily Safra Children’s Hospital, The Chaim Sheba Medical Center, Tel Hashomer 5265601, Israel; 4Physical Education Department, Kaye Academic College of Education, Beer Sheeva 8414201, Israel; 5Faculdade de Motricidade Humana, Universidade de Lisboa, Estrada da Costa, 1499-002 Cruz Quebrada-Dafundo, Portugal; amarques@fmh.ulisboa.pt; 6School of Education, Bar Ilan University, Ramat Gan 5290002, Israel; harelyo@biu.ac.il

**Keywords:** health-promoting schools, physical activity, dieting behavior, sedentary behavior

## Abstract

Schools with health-promoting school (HPS) frameworks are actively committed to enhancing healthy lifestyles. This study explored the contribution of school participation in HPS on students’ health behaviors, namely, physical activity (PA), sedentary behavior, and dieting. Data from the 2018/2019 Health Behavior in School-aged Children study on Israeli adolescents aged 11–17 years were used. Schools were selected from a sample of HPSs and non-HPSs. Between-group differences and predictions of health behavior were analyzed. No between-group differences were observed in mean number of days/week with at least 60 min of PA (HPS: 3.84 ± 2.19 days/week, 95% confidence interval of the mean = 3.02–3.34; non-HPS: 3.93 ± 2.17 days/week, 95% confidence interval of the mean = 3.13–3.38). Most children engaged in screen time behavior for >2 h/day (HPS: 60.83%; non-HPS: 63.91%). The odds of being on a diet were higher among more active children (odds ratio [OR] = 1.20), higher socio-economic status (OR = 1.23), and female (OR = 2.29). HPS did not predict any health behavior. These findings suggest that HPSs did not contribute to health behaviors more than non-HPSs. Therefore, health-promoting activities in HPSs need to be improved in order to justify their recognition as members of the HPS network and to fulfill their mission.

## 1. Introduction

Schools are recognized as significant settings in improving children’s health behaviors since a large proportion their day is spent there [1,2,3] and since schools virtually reach all children. The school environment plays a crucial role in providing opportunities for children to engage in health-promoting activities [1]. Therefore, the Health-Promoting School (HPS) framework, initiated by the World Health Organization (WHO), aims for a whole-school approach, with a focus on reorienting school systems toward sustainable health promotion [4]. The HPS framework focuses not only on classroom-based health education, but also on changes in school policy and the physical and social environment, using bottom-up involvement of pupils, parents, teachers, and staff. Bronfenbrenner’s ecological theory addresses the interaction of such factors at the environmental, organizational, and personal levels, thus fitting into the HPS framework. This theory is, therefore, able to adopt an approach that pertains to the whole school, from the students to specific classes, to school related activities. Nutrition-related policies and physical education classes also play a role within this theory; it has even been endorsed as a way to effectively promote better health behaviors in the school setting [5,6]. The ecological theory can address health-promoting behaviors at the different levels (environmental, organizational, personal) in an HPS framework, as the school setting encompasses all of these.

The HPS framework offers a holistic approach to organizational and systemic change for health and well-being in schools [7]. HPSs promote health through a multidisciplinary approach, promoting behaviors such as engagement in physical activity (PA), healthy eating, reduction of time spent in sedentary behaviors (mainly screen time), and abstinence from alcohol and tobacco consumption. This is important because these behaviors, individually or combined, are associated with adolescents’ health-related quality of life and are less subjective health complaints [8,9].

In parts of Australia, Canada, and the United Kingdom, efforts have been made to implement an HPS approach. Similar work has taken place in some schools in Hong Kong, Singapore, and Thailand over the last two decades [10]. Research has shown that an HPS approach may have a positive impact on students’ health promotion behaviors such as physical fitness and the intake of water, fruits, and vegetables, as well as their psychological well-being, oral health, and overall health outcomes [10,11,12,13].

In Israel, the Ministry of Education, in cooperation with the Ministry of Health, established an HPS network. The Israeli model is based on and a member of the WHO international network. Schools that voluntarily apply to join the network are committed to achieving educational objectives, promoting student health, and improving the quality of life and well-being of the school community—students, teachers and parents [14]. Candidates are required to declare their commitment to lead the school community towards implementation of a health promotion policy and to develop a multi-year program, integrated in the curriculum, aimed at forming healthy behavior and promoting a healthy lifestyle. Specifically, they are required to commit to a series of conditions including appointing a health coordinator, inclusion of the subject of health as an integral part of the school’s educational and social environment and values, commitment to healthy nutrition policies and educational program, and commitment to 90 min of additional physical activity a week in addition to the basic national requirements [15]. The program should be based on the standards for health promotion developed by the Ministry of Education and include goals, objectives, content, activities, follow-up, and evaluation that are consistent with the standards of the WHO [16]. Currently there are 1796 HPSs nationwide, 35% of the educational institutions from primary through high school, representing different ethnic and religious populations [17].

Despite the existing efforts of HPSs, many adolescents do not lead healthy lifestyles [18,19]. Schools may not be effective in promoting and consequently improving adolescents’ health. Studies have identified several barriers and challenges to HPS implementation and sustainability, including an over-emphasis on academic subjects, lack of institutional support [20], budgetary constraints, low prioritization of health initiatives, availability of unhealthy foods [21], lack of human resources, high workloads, time constraints [22], and structural and systemic barriers [23]. Thus, to understand the effectiveness of HPSs and compare their effectiveness in relation to schools that are not part of the HPS framework, the objective of the present study was to analyze the role of Israeli schools in promoting healthy behaviors, such as PA, healthy dieting, and reduced time spent on screen focused behaviors.

## 2. Materials and Methods

### 2.1. Study Design and Procedures

This study used Israeli data from the 2018–19 Health Behavior in School-aged Children (HBSC) WHO cross-national survey conducted among children aged 11–17. The HBSC is a school-based survey of adolescent health behaviors and psychosocial determinants carried out among representative samples of school-aged children every four years using an international standardized methodological protocol involving standardized procedures for sampling and translation of items. All measures in the HBSC study have been cross culturally validated. [24].

In order to ensure a representative sample, and in accordance with the international HBSC protocol [24], the Israeli Ministry of Education’s list of schools was used. HBSC international research protocol recommends a sample size of a minimum of 1500 students for each of the three age groups, in order to overcome the statistical dependence of children belonging to the same class. Additionally, this calculation ensures that a 95% confidence interval for the estimate of a proportion will have a maximum deviation of ±3 percentage points [25]. In 2018–2019, the sample in Israel included a total of 13,845 students, with the additional age groups. Sample weights were calculated to ensure relatively accurate representations of the subpopulations in the sample. Classrooms were randomly sampled (90% classroom response) and for each sampled school, an additional class was also randomly sampled. All students who were present in sampled classrooms were included (>95% pupil response). The research protocol received approval from ethics committees of the Israeli Ministry of Education and Bar Ilan University (No. 10,203).

### 2.2. Instruments and Measures

The HBSC survey is designed to measure key aspects of adolescents’ lives using validated questions that are relevant within a cross-national context. The questionnaire contains three types of questions: core items that are mandatory and are used to create the international datafile; optional packages that are not mandatory but allow more in-depth insights into specific topics while retaining comparability with at least some other countries; and national items that are country specific and may be responsive to national policy and funding priorities [26]. The research population included students from Hebrew speaking state-secular and state-religious schools. Participation in the study was fully voluntary and anonymous with no explicit incentives provided for participation. The survey is administered within the classroom setting using either paper or electronic questionnaires. Questionnaires were administrated by trained research assistants in the absence of a teacher during regular class time. Parents were informed about the study via the school administration and could opt out if they disagreed with it.

#### 2.2.1. Dependent Variables

Dieting behavior was assessed with the question, “Over the past year have you done any of the following things in order to lose weight or keep from gaining weight?” Response options included “No, my weight is fine”, “No, but I should lose some weight”, “No, because I need to put on weight”, and “Yes”. As the focus of this analysis was on dieting behavior rather than reasons for non-dieting, this variable was dichotomized into yes/no dieting for weight loss [27].

PA was measured by participants’ engagement in moderate-to-vigorous PA (MVPA). MVPA was defined as any activity that increased heart rate, as defined by the HBSC [24]. MVPA could include playing sports, running, walking, biking, or playing with friends. To calculate the amount of time spent on MVPA, the following question was asked: “Over the past 7 days, how many days were you physically active for a total of at least 60 min per day?” with response options ranging from 0–7 days. This variable was used as a continuous variable. In addition, as children and youth aged 5–17 should accumulate at least 60 min of MVPA daily [28], this variable was further used as a categorical variable (children who achieved and did not achieve the aforementioned PA recommendations).

Sedentary behavior was evaluated considering the overall sedentary activities index (used in the HBSC protocol since 1985/1986) that includes time devoted daily to watching television, playing computer games, and engaging in other screen time activities [29]. Response options were: “0 h/day”, “1 h/day”, “2 h/day”, “3 h/day”, “4 h/day”, and “5 h or more/day”. In addition, as the univariate structure of the scale supports combining the two questions into one indicator [29], total screen time score was also calculated. Screen time behavior was treated on an ordinal scale. In addition, as international guidelines recommend engaging in screen time behavior ≤2 h/day [30], the data were further categorized into children achieving and not achieving the aforementioned recommendations.

#### 2.2.2. Independent Variables

Independent variables included sex (male, female), grade (6th, 8th, and 10th–12th) belonging to HPS (yes or no). The Israeli Ministry of Education, in cooperation with the Ministry of Health, established a HPS network. Schools that agree to join in the network are committed to achieving educational objectives, in addition to promoting student health and improving the quality of life and well-being of the school community. The Israeli HPS is based on the international network approach and belongs to it [31].

The Family Affluence Scale (FAS) was used to assess socio-economic status. A sum score was constructed from the following four items: “Does your family own a car, van or truck?” Responses included: “No”, “Yes, one”, and “Yes, two or more” (0–2 points). “Do you have your own bedroom?” Responses included “No” or “Yes” (0–1 points). “During the past twelve months, how many times did you travel on holiday (vacation) with your family?” Responses included “Not at all”, “Once”, “Twice”, and “More than twice” (0–3 points). The final question was “How many computers does your family own?” Responses included “None”, “One”, “Two”, and “More than two” (0–3 points). The FAS score was divided into low (0–3 points), medium (4–6 points) and high FAS (7–9 points) [24].

### 2.3. Data Analysis

#### 2.3.1. School and Study Participants’ Characteristics

Descriptive statistics (mean, standard deviation, and prevalence) were used to describe the main school and participants’ characteristics of the total sample, HPSs and non-HPSs. To determine differences between HPSs and non-HPSs and participants’ characteristics, independent t-tests and chi-square tests were conducted.

#### 2.3.2. Between-Group Differences in Health Behavior

Between-group differences in health behavior, namely, the percentage of children achieving/not achieving PA level recommendations, engaging in screen behavior (television, computer, and total) for less or more than two h/day and dieting/not dieting in the past year, were examined via calculation of percentages and chi-squared tests. In addition, PA level was graphically presented using box plots with the central box representing the values from the lower to upper quartile (25th to 75th percentile); the vertical line extends from the minimum to the maximum value, excluding outside values, which are displayed as separate points. An outside value was defined as a value that was less than the lower quartile minus 1.5 times the interquartile range, or larger than the upper quartile plus 1.5 times the interquartile range. The middle line represents the median. Furthermore, between-group difference in PA level (number of days/week with at least 60 min of PA) was compared using independent-test and calculation of Cohen’s d effect size (mean ∆/standard deviation average from two means [24]). In general, values smaller than and equal to 0.20 were considered trivial effect sizes, values between 0.21 and 0.50 as small effect sizes, values 0.51–0.80 as moderate effect sizes, and values greater than 0.80 as large effect sizes [32].

#### 2.3.3. Variables Associated with and Predicting Health Behavior

Variables associated with health behavior were examined for each group with Pearson or Spearman correlations for the continuous variables (PA level and socio-economic status) and ordinal variables (screen time and grade), respectively. A Fisher’s *r*-to-*z* transformation was performed to test for potential differences between correlations. Prediction of health behavior was conducted for the entire sample with school type (HPS, non-HPS) as one of the predicting variables. Prediction of PA level (number of days with at least 60 min of PA) and total screen time (television screen time + computer screen time) was conducted using two separate forward multiple stepwise regression procedures. All independent variables were checked for multicollinearity using the variance of the inflation factor (VIF; VIF > 10; [33]). The criterion for inclusion in the model was an alpha level of 0.05, and the exclusion criterion was an alpha level of 0.10. Binary logistic regression modeling was used to determine the extent to which study variables, including school type (i.e., HPS and non-HPS), were predictive of dieting behavior (dieting in the past year). In that respect, the dependent variable (dieting behavior) was coded as 0, not dieting in the past year, and 1, dieting in the past year.

The data were analyzed with IBM SPSS statistics 19 (IBM, Armonk, NY, USA). In all statistical analyses, *p*-values lower than 0.05 were considered to indicate statistical significance.

## 3. Results

### 3.1. Schools’ and Study Participants’ Characteristics

There was a total of 1758 children from 64 schools. Thirty-eight percent (*n* = 669) of the children studied in HPSs (*n* = 28) and 62% (*n* = 1089) were from non-HPSs (*n* = 36). The prevalence of males and females in HPSs and non-HPSs was similar (prevalence of females: 49.50% and 44.80%, respectively; odds ratio = 0.79). However, in comparison to non-HPSs, in HPSs, the prevalence of younger children (6th and 8th grades) was significantly smaller (*p* < 0.001). In addition, in comparison to non-HPSs, the socio-economic status in the HPSs was lower (8.80 ± 2.42 vs. 8.24 ± 2.56, respectively). For additional information regarding school and study participant’s characteristics, see Table 1.

### 3.2. Between-Group Differences in Health Behavior

#### 3.2.1. Physical Activity Level

Engaging in at least 60 min of PA a day ranged from zero to seven days/week in both study groups; no between-group differences were observed in the mean number of days/week (HPSs: 3.84 ± 2.19 days/week, 95% confidence interval of the mean = 2.99–3.42; non-HPSs: 3.93 ± 2.17 days/week, 95% confidence interval of the mean = 3.13–3.38; *t*-statistic = −0.785, *p* = 0.432). Moreover, the between-group effect size was trivial (Cohen’s d = 0.04) (Figure 1). In both schools, 10% of the children met the recommended PA level; the prevalence of children who did not reach the recommended level was higher than that of children who met the recommended level (HPSs: chi-squared = 424.64, *p* < 0.001; non-HPSs: chi-squared = 696.64, *p* < 0.001). The between-group differences in prevalence of children meeting and not meeting the PA recommendations were non-significant (chi-squared = 0.00; *p* = 1.00). However, in both HPSs and non-HPSs, in comparison to females, males presented higher levels of PA (HPSs PA level—males: 4.24 ± 2.26, females: 3.34 ± 2.04, *t*-statistic = −4.83, *p* < 0.00001; non-HPSs PA level—males: 4.51 ± 2.23, females—3.45 ± 2.01, *t*-statistic = −8.22, *p* < 0.0001).

#### 3.2.2. Screen Time

In both HPSs and non-HPSs, most children (>60%) engaged in computer screen time or television screen time for up to two h/day. However, when computer and television screen times were combined, most children were engaged in screen time behavior for more than two h/day (HPSs: 60.83%; non-HPS: 63.91%). The between-group differences in prevalence of screen-time behavior were non-significant (chi-squared = 0.15, *p* < 0.05). In additional analyses, only total screen time was analyzed. In both school types, in comparison to females, males engaged in more screen time behavior (HPSs—males: 3.50 ± 2.42; females: 2.92 ± 2.12, *t* = −2.78, *p*-value = 0.007; non-HPSs—males: 3.88 ± 2.39, females: 2.01 ± 2.06, *t* = −5.44, *p* < 0.0001).

#### 3.2.3. Dieting Behavior

In both HPSs and non-HPSs, in comparison to the prevalence of children trying to lose weight in the past year, significantly more children were not trying to lose weight (66.80% and 68.50%, respectively; *p* < 0.05). The between-group differences in the prevalence of children trying to lose weight in the past year was non-significant (chi-squared = 0.75, *p* > 0.05). In both school types, in comparison to females, males engaged in less dieting behavior (HPSs—males: 0.16 ± 0.37; females: 0.33 ± 0.47, *t* = 4.23, *p* < 0.0001; non-HPSs—males: 0.20 ± 0.40, females: 0.27 ± 0.40, *t* = 2.41, *p* = 0.001).

#### 3.2.4. Between-Group Differences in Health Behavior

No between-group differences in health behavior (Table 2) were found.

### 3.3. Variables Associated with and Predicting Health Behavior

#### Associations between Study Variables and PA Level and Screen Time

Variables associated with health behavior were examined with Pearson or Spearman correlations for the continuous (PA level and socio-economic status) and ordinal (screen time and grade) variables, respectively. In both groups, grade was significantly negatively associated with PA level and screen time; older children were engaged less in PA and screen time behavior. However, Z transformation of the correlation test showed that the strength of the association between grade and PA level was significantly stronger in the non-HPSs (Z = 3.05; *p* < 0.001). In contrast, positive significant associations were found between socio-economic status and PA level in both schools; children from higher socio-economic status were engaged more in PA behavior. The strength of the association in both groups was similar (Z = −0.332; *p* > 0.05). In both schools, no significant associations were found between screen time and PA behavior (see Table 3).

As no between-group differences in health behavior were found (Table 2) and, overall, similar associations between study variables and health behaviors were observed (Table 3), prediction of health behavior was conducted jointly with both schools. A multiple regression analysis showed that higher levels of PA were predicted by less screen time, engaging in diet, lower grade (i.e., being younger), being a male, and having a higher socio-economic status. Being part of a HPS did not predict the level of PA. Overall, the model explained 15% of the observed PA level variability (F = 32.85; adjusted R^2^ = 0.15; *p* < 0.001). For additional information, refer to Table 4.

Screen time was predicted by PA level (active children engaged less in screen time), grade level (older children engaged less in screen time) and sex (females engaged less in screen time). Being part of an HPS did not predict the level of screen time behavior. Overall, the model explained four percent of the observed screen time behavior variability (F = 8.38; adjusted R^2^ = 0.04; *p* < 0.001). For additional information, refer to Table 5.

Finally, Table 6 shows a summary of multiple logistic regression analyses of the variables predicting dieting behavior. Hosmer–Lemeshow test showed that all variables were a good fit for selection into multiple logistic regression (*p* < 0.25) model. This final regression model was statistically significant (chi-squared = 59.52, *p* < 0.001) and explained 7% of the variance in dieting behavior (Nagelkerke R^2^ = 0.07). It was observed that the odds of being on diet were higher among children who were more active (odds ratio (OR) = 1.20, 95% confidence interval (CI) = 1.12–1.28, *p* < 0.001). In addition, in higher grades, there were greater odds of being on diet (OR = 1.23, 95% CI = 1.07–1.40, *p* = 0.002). Finally, the odds of being on diet among females were twice higher than for males (OR = 2.29, 95% CI = 1.71–3.07, *p* < 0.01). As with PA level and screen time, HPS did not predict dieting behavior (see Table 6).

## 4. Discussion

The main goal of the current study was to explore the contribution of school participation in the HPS framework with students’ characteristics: PA habits, screen time habits, and dieting behavior among Israeli schoolchildren aged 11–17 years. Our findings that affiliation with the HPS network did not contribute to students’ dieting behavior, screen time habits, or PA behaviors are supported by a systematic review conducted recently indicating that, although the WHO HPS network improved some aspects of students’ health, the effect of a school’s affiliation with the network was negligent [20]. The results of the current study indicate that individual characteristics, such as sex and grade, were associated with health behaviors.

In our study, there was no significant difference between HPS and non-HPS regarding engaging in 60 min of MVPA daily. Additionally, in both schools, only 10% of the children met the recommended PA level, a result that is consistent with the literature on PA habits of school-aged children [34,35,36]. Multiple logistic regression analyses showed that higher levels of PA were predicted by less screen time, engaging in diet, being younger, being a male, and higher socio-economic status. Sex and grade are known factors that affect the adoption of PA habits among adolescents [27,37,38,39]. Our finding that higher socio-economic status significantly predicted PA level is supported in several studies in adults, adolescents, and children [40,41,42]. Regarding the potential mechanism explaining the socio-economic status effect, Humbert et al. [40] reported that differences existed between adolescents from high compared to low socio-economic status regarding factors associated with PA participation. Those coming from families with higher socio-economic status reported less participation barriers. The impact of higher socio-economic status on PA should be considered among policymakers and specific programs should be developed, encouraging adolescents from families with lower socio-economic status to increase their participation in PA.

Our research indicated that most children in Israel, in both HPSs and non-HPSs, engaged in screen time more than the recommended guidance set by the WHO [2]. Similarly, the findings of the International Research Program on HBSC [24] reported that Israeli students ranked fourth among other countries in screen time use (using computers for play, internet, chat, email and homework). In addition, in the current study, multiple logistic regression analyses showed that screen time was predicted by PA level, grade, and sex. These results were not surprising, as grade and sex are known factors that affect the adoption of screen time habits among adolescents [27,37,38,39].

As evidenced by others [27,43] and further supported by our findings, engaging in more PA in adolescence is associated with increased odds of dieting behavior especially [24,43]. In the current study we also found that in comparison to males, the odds of being on diet was higher among females (OR = 2.29). Quick et al. [27] also found sex differences in dieting behavior. More specifically, it was reported that although there was an increasing trend in dieting among boys, a higher percentage of boys were overweight by medical standards than were actually dieting for weight loss. The prevalence of dieting among boys remained lower than for girls, suggesting that boys continued to be less concerned about their weight than girls. These findings are consistent with those from an international study among university students from 22 countries demonstrating that men were less aware they were overweight and less likely to diet to lose weight than women. The decreasing trend in dieting among non-overweight girls in this study is encouraging in terms of potentially leading to a lower incidence of disordered eating behaviors, but body dissatisfaction remains a common problem that could potentially spawn disordered eating and eating disorders [27]. Finally, the odds of being on diet were higher among children from higher socio-economic status (OR = 1.23). Manyanga et al. [44] showed in a multinational cross-sectional study among 9–11-year-old children (*n* = 6808) that when compared to participants in the highest socio-economic status groups, unhealthy diet pattern scores were significantly higher among those in the lowest within-country socio-economic status groups.

Quick et al. [27] found that younger students are more likely to report dieting behavior. This may be due to their lower level of independence compared with students in the higher grades, and thus, lower grade students could be less likely to be influenced by their friends and less exposed to peer pressure. This may indicate that health education influences students or, since this is a cross-sectional survey, it may suggest that those children with dieting behavior, screen time habits, and PA habits report higher levels of health education. Trigueros and colleagues found that if physical education classes stimulate learning via integrative methods, students will be more likely to participate in PA, as well as maintain healthy diet [43,45]. Previous research points to the importance of promoting self-initiating motivation to increase PA, in particular PA behavior outside of school. Knowledge transfer about PA may have a limited effect, therefore the HPS network may consider additional methods to promote PA self-motivation in addition to PE curriculum [46].

According to Bronfenbrenner’s bio-ecological model, there is an interaction between the environmental, organizational, and personal levels [47]; we found this to be relevant in our study, as we focused on students in the school setting. However, this interaction did not prove to show a difference between health-promoting behaviors between HPS and non-HPS schools.

Our study had a number of limitations. First, the design was based on cross-sectional data, in which baseline characteristics of HPSs and non-HPSs children significantly differed; thus, we were unable to establish temporality and causality. Second, repeated measures of the same variables and participants over long periods would be useful. Furthermore, findings were based on data gained from adolescents completing self-reported questionnaires; hence, future research should incorporate direct measurement (e.g., accelerometers). Lastly, the sample excluded adolescents who might have dropped out of school in high school.

## 5. Conclusions

The WHO’s concept of HPSs may be a promising approach in improving children’s lifestyles. In Israel there are specific commitments and additions to the curriculum and school environment in order to be formally recognized as HPS. However, in accordance with previous HPS studies, based on a large national sample of Israeli youth self-reporting their health behaviors, affiliation with the Israeli HPSs did not have a significant impact on students’ PA habits, sedentary behavior, or dietary habits. More specifically, in both schools, only 10% of the children met the recommended PA level, most children are engaged in screen time behavior for <2 h/day, and in comparison to the prevalence of children trying to lose weight in the past year, significantly more children were not trying to lose weight. Our research also indicated that students’ engagement in higher levels of PA were predicted by less screen time, engaging more in dieting behavior, being younger in age, from high socio-economic background, and being male. Sex was also related to differences in screen time and dietary behavior. It is possible that HPS in Israel have not fully adopted the different components of the WHO. In addition, there may be a need to adapt the international concept so that it better relates to the unique characteristics in the multicultural population in Israel. As a result, we recommend that intervention programs be tailored according to sex. In addition, policy changes and integrative programs in schools that engage the community, parents, and school staff may offer a more effective approach to health behavior promotion than relying on students only. Finally, in the future, it is important to formally monitor the HPS processes (e.g., number and types of activities provided and number of students participating in each activity).

Further research is needed to explore the association between the HPSs and students’ dieting behavior, screen time habits, and physical activities that takes into consideration additional variables such as family background, the social environment, and the actual health-promoting interventions performed at the schools.

## Figures and Tables

**Figure 1 ijerph-18-01183-f001:**
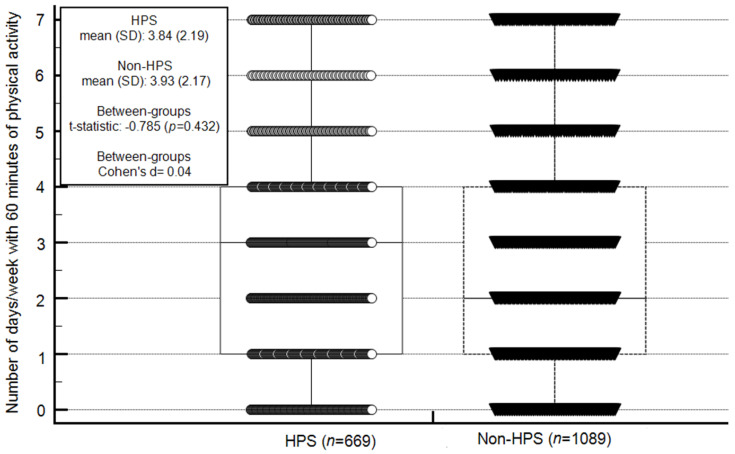
Between-group differences in number of days/week with 60 min of physical activity. Notes: HPS, health-promoting schools; Non-HPS, non-health-promoting schools; SD, standard deviation; in HPS and non-HPS, each circle or triangle, respectively, represents a child; The central box represents the lower to upper quartiles (25 to 75 percentile) of number of days/week in which children engaged in at least 60 min of physical activity; the vertical line extends from the minimum to maximum number of days/week with 60 min pf physical activity; The horizontal lines within the box represent the median number of days/week with 60 min of physical activity (HPS, 3 days/week, non-HPS, 2 days/week); no outside values were detected (an outside value was defined as a value that was smaller than the lower quartile minus 1.5 times the interquartile range, or larger than the upper quartile plus 1.5 times the interquartile range); the middle line represents the median.

**Table 1 ijerph-18-01183-t001:** Characteristics of schools and study participants.

Variable		Total (*n* = 1758): *n* (%) OR Mean (SD)	Health-Promoting School (*n* = 669): *n* (%) OR Mean (SD)	Non-Health-Promoting School (*n* = 1089): *n* (%) OR Mean (SD)	Between-Group Differences: Chi Square or *t*-Statistic (*p* Value)	Odds Ratio
Sex	Female, *n* (%)	932 (53.00)	331(49.50)	488 (44.80)	3.67 (0.06)	0.79
	Male, *n* (%)	826 (47.00)	338 (50.50)	601 (55.20)	3.67 (0.06)	
Grade	6th, *n* (%)	432 (24.60)	125 (18.70)	307 (28.20)	20.16 (<0.001)	1.70
	8th, *n* (%)	541 (30.80)	161 (24.10)	380 (34.90)	22.67 (<0.001)	1.69
	10th, *n* (%)	412 (23.40)	257 (38.40)	155 (14.20)	135.28 (<0.001)	0.26
	11th + 12th, *n* (%)	373 (21.20)	126 (18.80)	247 (22.70)	3.76 (0.06)	1.26
Weight, kg: mean (SD)		52.41 (14.11)	55.14 (14.12)	50.74 (13.85)	−6.16 (<0.001)	
Height, meters: mean (SD)		161.08 (12.41)	162.94 (12.95)	159.93 (11.94)	−4.83 (<0.001)	
Body mass index: mean (SD)		20.00 (3.95)	20.64 (4.04)	19.61 (3.84)	−5.08 (<0.001)	
Family affluence scale: mean (SD)		5.67 (1.92)	8.24 (2.56)	8.80 (2.42)	4.57 (<0.001)	

Notes: Body mass index was measured by kg/m^2^; SD, standard deviation.

**Table 2 ijerph-18-01183-t002:** Between-group differences in health behavior habits.

Variables		Health Promotion Schools (*n* = 669)		Non-Health Promotion Schools (*n* = 1089)		Between-Group Chi-Squared
		*n* (%)	Chi-Squared	*n* (%)	Chi-Squared	
Achieving physical activity level recommendations	Yes	68 (10.20)	424.64 *	109 (10.00)	696.64 *	0.44
	No	601 (89.80)		989 (90.00)		
Screen time—computer	<2 h/day	444 (66.36)	145.65 *	733 (67.30)	266.76 *	0.18
	>2 h/day	225 (33.63)		356 (32.69)		
Screen time—television	<2 h/day	434 (64.87)	112.45 *	714 (65.56)	209.23 *	0.18
	>2 h/day	235 (35.12)		375 (34.43)		
Screen time—total	<2 h/day	262 (39.16)	58.96 *	393 (36.08)	158.71 *	0.15
	>2 h/day	407 (60.83)		696 (63.91)		
Dieting in the past year	Yes	222 (33.18)	110.77 *	343 (31.49)	206.60 *	0.76
	No	447 (66.81)		746 (68.50)		

Notes: * significant differences in frequencies (*p* < 0.001; 2-tailed).

**Table 3 ijerph-18-01183-t003:** Continuous and ordinal variable associations with number of days/week with 60 min of physical activity and hours of screen time.

Variables	Number of Days with 60 min of Physical Activity			Hours of Screen Time		
	Health-Promoting School (*n* = 669): *n* (%) OR Mean (SD)	Non-Health-Promoting School (*n* = 1089): *n* (%) OR Mean (SD)	Z Transformation of the Correlation Test: Test Statistic Z	Health-Promoting School (*n* = 669): *n* (%) OR Mean (SD)	Non-Health-Promoting School (*n* = 1089): *n* (%) OR Mean (SD)	Z Transformation of the Correlation Test: Test Statistic Z
Grade	−0.091 *	−0.237 *	3.05 *	−0.120 *	−0.102 *	−0.37
Socio-economic status	0.137 *	0.153 *	−0.332	−0.074	0.0642	−0.356
Screen time	−0.072	−0.004	−1.38	----------------	----------------	----------------
Number of days with 60 min of physical activity	----------------	----------------	----------------	−0.045	0.002	−0.956

* Significant Spearman correlations or a significant test statistic z.; Abbreviation: SD, standard deviation.

**Table 4 ijerph-18-01183-t004:** Summary of multiple regression analysis: physical activity level.

Variables		Unstandardized Standard Error	Standardized Beta Coefficient	*t*	*p* Value
Constant		0.37		16.22	<0.001
Screen time, hours		0.05	−0.07	−2.90	0.004
Dieting behavior (reference: dieting)	Not dieting	0.12	−0.15	−5.826	<0.001
School sector (reference: secular)	Religious	0.06	−0.03	−1.22	0.22
	Arab	0.02	−0.02	−1.10	0.22
Grade		0.03	0.26	−9.71	<0.001
Sex (reference: female)	Male	0.12	0.24	9.25	<0.001
Socio-economic status		0.02	0.13	4.93	<0.001
Health-promoting school (reference: health-promoting school)	Non-health-promoting school	0.13	0.02	0.95	0.34
Model summery		F = 32.85; adjusted R^2^ = 0.15; *p* < 0.001			

Notes: Variance inflation factor in all analyses was <10.

**Table 5 ijerph-18-01183-t005:** Summary of multiple regression analysis: screen time.

Variables		Unstandardized Standard Error	Standardized Betta Coefficient	*t*	*p* Value
Constant		0.41		13.71	< 0.001
Physical activity level, days active		0.03	−0.08	−2.90	0.004
Dieting behavior (reference: dieting)	Not dieting	0.14	0.03	1.24	0.215
School sector (reference: secular)	Religious	0.07	0.00	−0.94	0.925
	Arab	0.06	0.00	−0.98	0.987
Grade		0.06	−0.12	−3.97	< 0.001
Sex (reference: female)	Male	0.13	0.19	6.61	< 0.001
Socio-economic status		0.02	−0.01	−0.31	0.756
Health-promoting school (reference: health-promoting school)	Non-health-promoting school	0.13	−0.01	−0.63	0.528
Model summary		F = 8.38; adjusted R^2^ = 0.04; *p* < 0.001			

Notes: Variance inflation factor in all analyses was <10.

**Table 6 ijerph-18-01183-t006:** Summary of multiple binary logistic regression analysis: dieting behavior.

Variables		Odds Ratio	Wald	95% Confidence Interval	*p* Value
Constant			54.42		<0.001
Physical activity level, days/week		1.20	32.69	1.12–1.28	<0.001
Screen time, hours		1.03	1.36	0.97–1.10	0.24
School sector (reference, secular)	Religious	0.92	0.27	0.67–1.25	0.60
	Arab	0.84	0.70	0.56–1.26	0.40
Grade		1.23	9.38	1.07–1.40	0.002
Sex (reference male)	Female	2.29	31.57	1.71–3.07	<0.001
Socio-economic status		1.00	0.08	0.95–1.06	0.76
Health-promoting school (reference: health-promoting school)	Non-health-promoting school	1.03	0.05	0.77–1.38	0.81
Model summary		Chi-squared = 59.52, *p* < 0.001, Nagelkerke R^2^ = 0.07.			

Note: dieting behavior reference: no dieting.

## Data Availability

The data presented in this study are available on request from the corresponding author. The data are not publicly available due to ethical considerations of the ministry of health.

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
