# Peer review of "Can Health-Promoting Schools Contribute to Better Health Behaviors? Physical Activity, Sedentary Behavior, and Dietary Habits among Israeli Adolescents"

_ijerph, 2021, doi:10.3390/ijerph18031183_

Round 1

Reviewer 1 Report

The manuscript under review provides an assessment of whether indices of lifestyle health behaviors are improved in Health Promoting Schools over those schools not participating in the WHO framework using a national sample of Israeli youth.   The minor concerns are listed below:

Minor

Figure 1.   is confusing.  The mean values are bold, and close to values of 4, yet are placed between values of 2 and 3.  —It isn’t clear whether the mean values are incorrect.   Moreover, it is a bit odd to use a figure for these data when the most important result is that only 10% of students reach the recommended PA level. 

To improve the clarity of the manuscript — suggest Results,Tables, and Figure should all be consistent with first/second and left/right presentation of HPC and non-HPC.  

To improve clarity of manuscript—suggest tables providing the multiple regression predictions indicate the collapsed group nature of the analysis in the table title or legend. 

Line 284  edit:  replace “reaching” with reach

Author Response

Dear Reviewer

Thank you very much for the significant comments

 Review Report Form 1

The manuscript under review provides an assessment of whether indices of lifestyle health behaviors are improved in Health Promoting Schools over those schools not participating in the WHO framework using a national sample of Israeli youth.   The minor concerns are listed below:

Minor

Figure 1.   is confusing.  The mean values are bold, and close to values of 4, yet are placed between values of 2 and 3.  —It isn’t clear whether the mean values are incorrect.   Moreover, it is a bit odd to use a figure for these data when the most important result is that only 10% of students reach the recommended PA level.

The mean value written in the figure is correct. However, indeed, its location is confusing. Therefore, we moved the means and standard deviations from the box plot itself to the figure's upper left corner in which the mean values are compared using t-test. We selected this figure in order to provide the reader with additional descriptive statistics and to give a better appreciation of the distribution of participants' physical activity level.  

To improve the clarity of the manuscript — suggest Results,Tables, and Figure should all be consistent with first/second and left/right presentation of HPC and non-HPC.

The required corrections were conducted in Figure 1. The Results section was also reviewed and corrected accordingly. As in the Tables the HPSs appeared first, no additional corrections were made to the tables.  

 To improve clarity of manuscript—suggest tables providing the multiple regression predictions indicate the collapsed group nature of the analysis in the table title or legend. 

A row was added to table 4b in which the model's F, adjusted R2, and p values are written.

Line 284  edit:  replace “reaching” with reach

Thank you we replace the word reaching

Reviewer 2 Report

Please edit the tables marked - the first two columns are very hard to make out what you are wanting the reader to see and understand. Also, see note about missing word in the abstract.

Very well written, good analysis, interesting to learn about HPS in Israel.

Author Response

Dear Reviewer

Thank you very much for the significant comments

Review Report Form 2

Please edit the tables marked - the first two columns are very hard to make out what you are wanting the reader to see and understand. Also, see note about missing word in the abstract.

Tables were reedited. More specifically, the variables' names were moved from the center to the left, the names of the variables are now in the same line as the numbers, and spaces between variables were added.  We also made the required corrections in the abstract.

Very well written, good analysis, interesting to learn about HPS in Israel.

Thank you so much!

Reviewer 3 Report

The paper is interesting and well documented. It approches  a very  important  aspect  of the  lifestyle in children and adolescents. 

The results  reported  are relevant , however some  data  regarding  the  lifestyle  habits of parents  and family colud  improve the  main message and the  significance   of  the manuscript's  conclusions.  

In addition some hydration status ( TBW,ECW,ICW, Phase Angle)  data coulb  be  relevant to  implement the  life style investigation .

Author Response

Review Report Form 3

The paper is interesting and well documented. It approaches a  very  important  aspect  of the  lifestyle in children and adolescents. The results reported are relevant, however some data regarding  the  lifestyle  habits of parents  and family colud  improve the  main message and the  significance   of  the manuscript's  conclusions.  In addition some hydration status (TBW,ECW,ICW, Phase Angle)  data could  be  relevant to  implement the  life style investigation .

We agree that parental and familial lifestyle habits could improve the message of the paper, though this was not within the scope of the current paper. Future studies should be focused on these aspects.

Reviewer 4 Report

1. The Introduction section failed to convincingly rationalize the importance and relevance of the topic of this manuscript in relation to the intended scope and content of this journal's work. In this sense, although I appreciate the authors' elaborations on the justification of the study, a striking issue points to the lack of a theoretical description that would cohere the different aspects dealt with in this section.
It would have been useful if the authors had decided to borrow the justifications belonging to the self-determination theory (1-3,7) and the trans-contextual model [3-7].
2. The study also did not elaborate a solid theoretical reason for formulating the hypotheses of the importance of physical education classes towards the promotion of physical sports practice.
Methodology
3. There were many missing information in the Participants section. For instance, in which country or context did the authors recruit sample? What was their basis for determining this sample size? Did they conduct power analysis? What was the profile of the participants in terms of their year level and socioeconomic status? These details are important to provide a basis for interpreting generalizability of the study's findings.
4. A more complete description of the instrument used in this study is needed.
5. Another fundamental element which concerns me enormously is knowing whether or not there are any initial differences in the study between the Health promoting group and the Non-health promoting group, since if there were, this would have a great influence on the results of the study.
Discussion
6. According to comment number 1, the authors should restructure this section. Furthermore, it is necessary to establish comparisons between the results obtained in their study and other similar existing ones.
References
36 references are insufficient for an experimental study, I hope that my comments will help to increase the number of references.

References

1.Ntoumanis, N. (2001). A self‐determination approach to the understanding of motivation in physical education. British journal of educational psychology, 71(2), 225-242.   2.Standage, M., Duda, J. L., & Ntoumanis, N. (2005). A test of self‐determination theory in school physical education. British Journal of Educational Psychology, 75(3), 411-433.   3. Trigueros, R., Mínguez, L. A., González-Bernal, J. J., Jahouh, M., Soto-Camara, R., & Aguilar-Parra, J. M. (2019). Influence of teaching style on physical education adolescents’ motivation and health-related lifestyle. Nutrients, 11(11), 2594.   4. Koka, A., Tilga, H., Kalajas-Tilga, H., Hein, V., & Raudsepp, L. (2020). Detrimental effect of perceived controlling behavior from physical education teachers on students’ leisure-time physical activity intentions and behavior: An application of the Trans-Contextual Model. International Journal of Environmental Research and Public Health, 17(16), 5939.   5.Wang, Y., & Chen, A. (2020). Two pathways underlying the effects of physical education on out-of-school physical activity. Research Quarterly for Exercise and Sport, 91(2), 197-208.   6. Hagger, M. S., Chatzisarantis, N. L., Culverhouse, T., & Biddle, S. J. (2003). The processes by which perceived autonomy support in physical education promotes leisure-time physical activity intentions and behavior: a trans-contextual model. Journal of educational psychology, 95(4), 784.   7. Trigueros, R., Aguilar-Parra, J. M., Cangas, A. J., López-Liria, R., & Álvarez, J. F. (2019). Influence of physical education teachers on motivation, embarrassment and the intention of being physically active during adolescence. International journal of environmental research and public health, 16(13), 2295.

Author Response

Dear Reviewer

Thank you very much for the significant comments

Review Report Form 4

  1. The Introduction section failed to convincingly rationalize the importance and relevance of the topic of this manuscript in relation to the intended scope and content of this journal's work. In this sense, although I appreciate the authors' elaborations on the justification of the study, a striking issue points to the lack of a theoretical description that would cohere the different aspects dealt with in this section.
    It would have been useful if the authors had decided to borrow the justifications belonging to the self-determination theory (1-3,7) and the trans-contextual model [3-7].

We believe that Bronfenbrenner’s ecological theory best fit our study. We have added the following text to the introduction: Bronfenbrenner’s ecological theory addresses the interaction of such factors at the environmental, organizational, and personal levels, therefore fitting into the HPS framework. This theory is therefore able to adopt an approach that pertains to the whole school, from the students to specific classes, to school related activities. Nutrition-related policies and physical education classes also play a role within this theory; it has even been endorsed as a way to effectively promote better health behaviors in the school setting [5,6]. The ecological theory can address health promoting behaviors at the different levels (environmental, organizational, personal) in an HPS framework, as the school setting encompasses all of these.

We also added this following section to the discussion:

According to Bronfenbrenner’s ecological model, there is an interaction between the environmental, organizational, and personal levels; we found this to be relevant in our study, as we focused on students in the school setting. However, this interaction did not prove to show a difference between health promoting behaviors between HPS and non-HPS schools.

  1. The study also did not elaborate a solid theoretical reason for formulating the hypotheses of the importance of physical education classes towards the promotion of physical sports practice.

Methodology
3. There were many missing information in the Participants section. For instance, in which country or context did the authors recruit sample? What was their basis for determining this sample size? Did they conduct power analysis? What was the profile of the participants in terms of their year level and socioeconomic status? These details are important to provide a basis for interpreting generalizability of the study's findings.

As previously noted in the methodology section the study is based on Israeli data using a list of schools from the Ministry of Education to sample classrooms. All students in sampled classrooms present were included (> 95% pupil response). We have added further information regarding the basis for the sample size. The sample is designed to represent the different subpopulations in Israel, including year level and socioeconomic status, as indicated in the paper.

  1. A more complete description of the instrument used in this study is needed.

We have added further information on the survey instrument.

  1. Another fundamental element which concerns me enormously is knowing whether or not there are any initial differences in the study between the Health promoting group and the Non-health promoting group, since if there were, this would have a great influence on the results of the study.

Differences between HPS and non-HPS in school in characteristics and study participants are listed in table 1. The prevalence of males and females in HPSs and non-HPSs was similar. However, in comparison to non-HPSs, in HPSs, the prevalence of younger children (6th and 8th grades) was significantly smaller. In addition, in comparison to non-HPSs, the socio-economic status in the HPSs was lower (8.80+2.42 vs. 8.24+2.56, respectively). This information appears in the first section of the results: 3.1 Schools’ and study participants' characteristics. This information was also added to the limitations section.

Discussion
6. According to comment number 1, the authors should restructure this section. Furthermore, it is necessary to establish comparisons between the results obtained in their study and other similar existing ones.

References
36 references are insufficient for an experimental study, I hope that my comments will help to increase the number of references.

Thank you

References

  1. Trigueros, R., Mínguez, L. A., González-Bernal, J. J., Jahouh, M., Soto-Camara, R., & Aguilar-Parra, J. M. (2019). Influence of teaching style on physical education adolescents’ motivation and health-related lifestyle. Nutrients11(11), 2594.
  2. Koka, A., Tilga, H., Kalajas-Tilga, H., Hein, V., & Raudsepp, L. (2020). Detrimental effect of perceived controlling behavior from physical education teachers on students’ leisure-time physical activity intentions and behavior: An application of the Trans-Contextual Model. International Journal of Environmental Research and Public Health17(16), 5939.

5.Wang, Y., & Chen, A. (2020). Two pathways underlying the effects of physical education on out-of-school physical activity. Research Quarterly for Exercise and Sport91(2), 197-208.

  1. Hagger, M. S., Chatzisarantis, N. L., Culverhouse, T., & Biddle, S. J. (2003). The processes by which perceived autonomy support in physical education promotes leisure-time physical activity intentions and behavior: a trans-contextual model. Journal of educational psychology95(4), 784.
  2. Trigueros, R., Aguilar-Parra, J. M., Cangas, A. J., López-Liria, R., & Álvarez, J. F. (2019). Influence of physical education teachers on motivation, embarrassment and the intention of being physically active during adolescence. International journal of environmental research and public health16(13), 2295.

Reviewer 5 Report

Does the introduction provide sufficient background and include all relevant references?

Studies carried out in Latin America can be included, more information about this can be found in the following study (Bastos et al., 2020).  Studies conducted in the Pacific can also be included (Xu et al., 2020).

On the other hand, I believe that this bibliographical reference is indispensable and has not been included (Langford et al., 2014).

Is the research design appropriate?

I think the research design is appropriate.

Are the methods adequately described?

Include the code of ethics committee.

The reliability and validity of all the instruments used in the study should be included, both in the original version and in the Hebrew version.

Are the results clearly presented?

It is very important to present the data in tables 1,2 and 3 disaggregated by sex, as well as the data in figure 1.

In the discussion, it would be interesting to inquire into the explanation of why Israeli boys do more physical activity than girls. It would also be interesting to make specific proposals for the WHO program to be successful in Israel.

Are the conclusions supported by the results?

I think the conclusions are based on the results found.

References

Bastos, P. O., Cavalcante, A. S. P., Pereira, W. M. G., de Castro, V. H. S., Ferreira Junior, A. R., Guerra, P. H., . . . Barbosa Filho, V. C. (2020). Health Promoting School Interventions in Latin America: A Systematic Review Protocol on the Dimensions of the RE-AIM Framework. Int J Environ Res Public Health, 17(15). doi:10.3390/ijerph17155558

Langford, R., Bonell, C. P., Jones, H. E., Pouliou, T., Murphy, S. M., Waters, E., . . . Campbell, R. (2014). The WHO Health Promoting School framework for improving the health and well-being of students and their academic achievement. Cochrane Database Syst Rev(4), CD008958. doi:10.1002/14651858.CD008958.pub2

Xu, T., Tomokawa, S., Gregorio, E. R., Jr., Mannava, P., Nagai, M., & Sobel, H. (2020). School-based interventions to promote adolescent health: A systematic review in low- and middle-income countries of WHO Western Pacific Region. PLoS One, 15(3), e0230046. doi:10.1371/journal.pone.0230046

Author Response

Dear Reviewer

Thank you very much for the significant comments

Review Report Form 5

Does the introduction provide sufficient background and include all relevant references?

Studies carried out in Latin America can be included, more information about this can be found in the following study (Bastos et al., 2020).  Studies conducted in the Pacific can also be included (Xu et al., 2020).

On the other hand, I believe that this bibliographical reference is indispensable and has not been included (Langford et al., 2014).

Thank you for these additional references. Please note the Langford study was included in the initial submission and we have added additional background information in the introduction section and discussion related to both theory and previous experience in HPS networks.

Is the research design appropriate?

I think the research design is appropriate.

Are the methods adequately described?

Include the code of ethics committee.

Thank you we added the approval code of the ethics committee.

The reliability and validity of all the instruments used in the study should be included, both in the original version and in the Hebrew version.

We have added reference to the validity of the measures used in the HBSC study.

Are the results clearly presented?

It is very important to present the data in tables 1,2 and 3 disaggregated by sex, as well as the data in figure 1.

We understand the importance of presenting the results in the tables and in the figure also by sex. However, as we were limited in number of tables and figures and in order to not put too much information in the tables and making them difficult to understand, we decided to add the information regarding between-sex differences to the text itself and not to the tables and figures. Accordingly, differences in PA behavior, screen-time behavior and dieting behavior, can be found in sections 3.2.1, 3.2.2, and 3.2.3, respectively.

In the discussion, it would be interesting to inquire into the explanation of why Israeli boys do more physical activity than girls. It would also be interesting to make specific proposals for the WHO program to be successful in Israel.

Thank you

Are the conclusions supported by the results?

I think the conclusions are based on the results found.

References

Bastos, P. O., Cavalcante, A. S. P., Pereira, W. M. G., de Castro, V. H. S., Ferreira Junior, A. R., Guerra, P. H., . . . Barbosa Filho, V. C. (2020). Health Promoting School Interventions in Latin America: A Systematic Review Protocol on the Dimensions of the RE-AIM Framework. Int J Environ Res Public Health, 17(15). doi:10.3390/ijerph17155558

Langford, R., Bonell, C. P., Jones, H. E., Pouliou, T., Murphy, S. M., Waters, E., . . . Campbell, R. (2014). The WHO Health Promoting School framework for improving the health and well-being of students and their academic achievement. Cochrane Database Syst Rev(4), CD008958. doi:10.1002/14651858.CD008958.pub2

Xu, T., Tomokawa, S., Gregorio, E. R., Jr., Mannava, P., Nagai, M., & Sobel, H. (2020). School-based interventions to promote adolescent health: A systematic review in low- and middle-income countries of WHO Western Pacific Region. PLoS One, 15(3), e0230046. doi:10.1371/journal.pone.0230046

Reviewer 6 Report

This cross-sectional study aimed to investigate the role of Schools with Health Promoting School (HPS) frameworks in Israeli schools in promoting healthy behaviors, such as PA, healthy dieting, and sedentary behaviours and compare their effectiveness in relation to schools that are not part of the HPS framework. It is very interesting approach. But the results of the study are minimal in this regard.  Writers could even consider the change the main research question for ‘Physical Activity, Sedentary Behavior, and Dietary Habits among Israeli Adolescents’

Major revision

The main problem is that the reader do not know what is done at schools with HPS framework. In the introduction there has told very general level that HPS schools try to promote healthy behaviours with a holistic approach to organisational and systemic change in schools. In this article there is a need to know what those Israeli schools has done. It is very natural that the results of this study are not different comparing to non-HPS schools if joining the WHO framework is enough without any actions.  

I don’t either understand that why to list other countries participation HPS without study results in healthy behaviours. Instead, there are earlier identified barriers and challenges to HPS implementation and sustainability e.g. over-emphasis on academic subjects, lack of institutional support etc. But nothing were done for these barriers in this study, they were not used in study. Also, the framework and the barriers was not discussed at discussion part of the study. The discussion remains insufficient because the schools promoting programs and actions are not known and not dealt in discussion.

The main results are actually the correlates of the healthy behaviour in Israeli adolescents.   

Minor revisions

Abstract

line 23 Sedentary behaviour

line 30 I like to see with OR also CI

line 31-33 Conclusion is ok put very general ‘health promoting activities in HPSs need to be improved’   when the reader could not find what has done and what to improve.

Methods

In the methods section, the design and overall setting are clearly presented. The sample size is quite large. Variables are clearly presented. But I like to know

What kind of actions has done at HPS schools?

Dieting is not always a healthy behaviour. There is need for ‘open’ the concept.

Result

Table 1, where are ORs?

Figure 1 is unclear, what means the colours, what is central box?

Variables associated with and predicting health behaviour could be presented in uncomplicated OR+CI table

Discussion

In this study, there is well discussed and compared to other studies that higher levels of PA were predicted by less screen time, engaging in diet, being younger, being a male, and higher socio-economic status and that higher socio-economic status significantly predicted PA . Also, the potential mechanism explaining these results are discussed well. For screentime and dieting the situation is same, discussion is ok.

The conclusion part repeats the earlier discussion.

The writers did not discussed of the result of the main study question. This is not enough:   “affiliation with the HPS network did not contribute to students’ dieting behavior, screen time habits, or PA behaviors are supported  by a systematic review conducted recently indicating that, although the WHO HPS network improved some aspects of students' health, the effect of a school’s affiliation with the network was negligent [14]”

Author Response

Dear Reviewer

Thank you very much for the significant comments

Review Report Form 6

This cross-sectional study aimed to investigate the role of Schools with Health Promoting School (HPS) frameworks in Israeli schools in promoting healthy behaviors, such as PA, healthy dieting, and sedentary behaviours and compare their effectiveness in relation to schools that are not part of the HPS framework. It is very interesting approach. But the results of the study are minimal in this regard.  Writers could even consider the change the main research question for ‘Physical Activity, Sedentary Behavior, and Dietary Habits among Israeli Adolescents’

Major revision

The main problem is that the reader do not know what is done at schools with HPS framework. In the introduction there has told very general level that HPS schools try to promote healthy behaviours with a holistic approach to organisational and systemic change in schools. In this article there is a need to know what those Israeli schools has done. It is very natural that the results of this study are not different comparing to non-HPS schools if joining the WHO framework is enough without any actions.  

We have added details on the requirements for participation in the HPS framework in the introduction. There are specific commitments and additions to the curriculum and school environment in order to be formally recognized as HPS.

I don’t either understand that why to list other countries participation HPS without study results in healthy behaviours. Instead, there are earlier identified barriers and challenges to HPS implementation and sustainability e.g. over-emphasis on academic subjects, lack of institutional support etc. But nothing were done for these barriers in this study, they were not used in study. Also, the framework and the barriers was not discussed at discussion part of the study. The discussion remains insufficient because the schools promoting programs and actions are not known and not dealt in discussion.

The main results are actually the correlates of the healthy behaviour in Israeli adolescents.   

Minor revisions

Abstract

line 23 Sedentary behavior

Thank you we correct the missing word

line 30 I like to see with OR also CI

We apologize if we did not understand you correctly, but we did not add ORs as the variable of interest is not dichotomous. We did add the 95% CI of the mean. We added this information also to the results section (3.2.1 Physical activity level)

line 31-33 Conclusion is ok put very general ‘health promoting activities in HPSs need to be improved’   when the reader could not find what has done and what to improve.

Methods

In the methods section, the design and overall setting are clearly presented. The sample size is quite large. Variables are clearly presented. But I like to know

What kind of actions has done at HPS schools?

Thank you we added the information regarding the actions that has done at HPS

Dieting is not always a healthy behaviour. There is need for ‘open’ the concept.

Result

Table 1, where are ORs?

ORs for all categorical variables were added to the table.

Figure 1 is unclear, what means the colours, what is central box?

We understand that the mean values that originally were written in the box plot itself might have been confusing. Therefore, the mean values were moved to the upper left corner of the figure. The central box represents the values from the lower to upper quartiles (25 to 75 percentile); the vertical line extends from the minimum to maximum value; the middle line represents the median. This information is written under the figure, in the figure's notes. The only colors in the figures are black and white. It is possible that in places in which there are a lot of participants and therefore a lot of circles, because of the density of the circles, the color might appear different (i.e., darker). 

Variables associated with and predicting health behaviour could be presented in uncomplicated OR+CI table

Our sincere apologize; we don't fully understand your comment. To our best knowledge, it is acceptable to present association tables using person or spearman correlations. Moreover, we did address between-group differences in correlations strength using Z transformation of the correlation test. 

Discussion

In this study, there is well discussed and compared to other studies that higher levels of PA were predicted by less screen time, engaging in diet, being younger, being a male, and higher socio-economic status and that higher socio-economic status significantly predicted PA . Also, the potential mechanism explaining these results are discussed well. For screentime and dieting the situation is same, discussion is ok.

The conclusion part repeats the earlier discussion.

The writers did not discussed of the result of the main study question. This is not enough:  “affiliation with the HPS network did not contribute to students’ dieting behavior, screen time habits, or PA behaviors are supported  by a systematic review conducted recently indicating that, although the WHO HPS network improved some aspects of students' health, the effect of a school’s affiliation with the network was negligent [14]”

Thank you for your comment, we have expanded the discussion and conclusions in response to the main study question.

Round 2

Reviewer 4 Report

Thank you for all changed, but You should add more international studies that reinforce the role of PE in supporting active physical activity habits.

Trigueros, R., Mínguez, L. A., González-Bernal, J. J., Jahouh, M., Soto-Camara, R., & Aguilar-Parra, J. M. (2019). Influence of teaching style on physical education adolescents’ motivation and health-related lifestyle. Nutrients11(11), 2594.

Koka, A., Tilga, H., Kalajas-Tilga, H., Hein, V., & Raudsepp, L. (2020). Detrimental effect of perceived controlling behavior from physical education teachers on students’ leisure-time physical activity intentions and behavior: An application of the Trans-Contextual Model. International Journal of Environmental Research and Public Health17(16), 5939.

Wang, Y., & Chen, A. (2020). Two pathways underlying the effects of physical education on out-of-school physical activity. Research Quarterly for Exercise and Sport91(2), 197-208.

Hagger, M. S., Chatzisarantis, N. L., Culverhouse, T., & Biddle, S. J. (2003). The processes by which perceived autonomy support in physical education promotes leisure-time physical activity intentions and behavior: a trans-contextual model. Journal of educational psychology95(4), 784.

Trigueros, R., Aguilar-Parra, J. M., Cangas, A. J., López-Liria, R., & Álvarez, J. F. (2019). Influence of physical education teachers on motivation, embarrassment and the intention of being physically active during adolescence. International journal of environmental research and public health16(13), 2295

Author Response

Thank you for this suggestion. We have added international studies that speak to the role of PE in supporting PA habits, including Trigueros et al. and Wang & Chen.

Trigueros, R., Mínguez, L. A., González-Bernal, J. J., Jahouh, M., Soto-Camara, R., & Aguilar-Parra, J. M. (2019). Influence of teaching style on physical education adolescents’ motivation and health-related lifestyle. Nutrients11(11), 2594.

Koka, A., Tilga, H., Kalajas-Tilga, H., Hein, V., & Raudsepp, L. (2020). Detrimental effect of perceived controlling behavior from physical education teachers on students’ leisure-time physical activity intentions and behavior: An application of the Trans-Contextual Model. International Journal of Environmental Research and Public Health17(16), 5939.

Wang, Y., & Chen, A. (2020). Two pathways underlying the effects of physical education on out-of-school physical activity. Research Quarterly for Exercise and Sport91(2), 197-208.

Hagger, M. S., Chatzisarantis, N. L., Culverhouse, T., & Biddle, S. J. (2003). The processes by which perceived autonomy support in physical education promotes leisure-time physical activity intentions and behavior: a trans-contextual model. Journal of educational psychology95(4), 784.

Trigueros, R., Aguilar-Parra, J. M., Cangas, A. J., López-Liria, R., & Álvarez, J. F. (2019). Influence of physical education teachers on motivation, embarrassment and the intention of being physically active during adolescence. International journal of environmental research and public health16(13), 2295

Reviewer 5 Report

I think the article can be published after the improvements.

Author Response

Thank you we did all the improvements that we ask for.

Reviewer 6 Report

I think Figure 1 is still unclear, text will be enough.

Author Response

Review 6

I think Figure 1 is still unclear, text will be enough.

Thank you, we have added the following description to Figure 1’s caption:

Notes: HPS, health promoting schools; Non-HPS, non-health promoting schools; SD, standard deviation; in HPS and non-HPS, each circle or triangle, respectively, represents a child; The central box represents the lower to upper quartiles (25 to 75 percentile) of number of days/week in which children engaged in at least 60 minutes of physical activity; the vertical line extends from the minimum to maximum number of days/week with 60 minutes pf physical activity; The horizontal lines within the box represent the median number of days/week with 60 minutes of physical activity (HPS, 3 days/week, non-HPS, 2 days/week); no outside values were detected (an outside value was defined as a value that was smaller than the lower quartile minus 1.5 times the interquartile range, or larger than the upper quartile plus 1.5 times the interquartile range); the middle line represents the median.

This manuscript is a resubmission of an earlier submission. The following is a list of the peer review reports and author responses from that submission.